# Towards Accurate Point-of-Care Tests for Tuberculosis in Children

**DOI:** 10.3390/pathogens11030327

**Published:** 2022-03-08

**Authors:** Nina Vaezipour, Nora Fritschi, Noé Brasier, Sabine Bélard, José Domínguez, Marc Tebruegge, Damien Portevin, Nicole Ritz

**Affiliations:** 1Mycobacterial and Migrant Health Research Group, University Children’s Hospital Basel, Department for Clinical Research, University of Basel, 4056 Basel, Switzerland; nina.vaezipour@ukbb.ch (N.V.); nora.fritschi@ukbb.ch (N.F.); 2Infectious Disease and Vaccinology Unit, University Children’s Hospital Basel, University of Basel, 4056 Basel, Switzerland; 3Department of Health Sciences and Technology, Institute for Translational Medicine, ETH Zurich, 8093 Zurich, Switzerland; noekarl.brasier@usb.ch; 4Department of Digitalization & ICT, University Hospital Basel, 4031 Basel, Switzerland; 5Department of Pediatric Respiratory Medicine, Immunology and Critical Care Medicine, Charité–Universitätsmedizin Berlin, 13353 Berlin, Germany; sabine.belard@charite.de; 6Institute of Tropical Medicine and International Health, Charité–Universitätsmedizin Berlin, 13353 Berlin, Germany; 7Institute for Health Science Research Germans Trias i Pujol. CIBER Enfermedades Respiratorias, Universitat Autònoma de Barcelona, 08916 Barcelona, Spain; jadominguez@igtp.cat; 8Department of Infection, Immunity and Inflammation, UCL Great Ormond Street Institute of Child Health, University College London, London WCN1 1EH, UK; m.tebruegge@ucl.ac.uk; 9Department of Pediatrics, The Royal Children’s Hospital Melbourne, The University of Melbourne, Parkville, VIC 3052, Australia; 10Department of Medical Parasitology and Infection Biology, Swiss Tropical and Public Health Institute, 4123 Allschwil, Switzerland; damien.portevin@swisstph.ch; 11University of Basel, 4001 Basel, Switzerland; 12Department of Paediatrics and Paediatric Infectious Diseases, Children’s Hospital, Lucerne Cantonal Hospital, 6000 Lucerne, Switzerland

**Keywords:** GeneXpert, LAM, LAMP, lipoarabinomannan, POCT, POCUS, sonography, Truenat

## Abstract

In childhood tuberculosis (TB), with an estimated 69% of missed cases in children under 5 years of age, the case detection gap is larger than in other age groups, mainly due to its paucibacillary nature and children’s difficulties in delivering sputum specimens. Accurate and accessible point-of-care tests (POCTs) are needed to detect TB disease in children and, in turn, reduce TB-related morbidity and mortality in this vulnerable population. In recent years, several POCTs for TB have been developed. These include new tools to improve the detection of TB in respiratory and gastric samples, such as molecular detection of Mycobacterium tuberculosis using loop-mediated isothermal amplification (LAMP) and portable polymerase chain reaction (PCR)-based GeneXpert. In addition, the urine-based detection of lipoarabinomannan (LAM), as well as imaging modalities through point-of-care ultrasonography (POCUS), are currently the POCTs in use. Further to this, artificial intelligence-based interpretation of ultrasound imaging and radiography is now integrated into computer-aided detection products. In the future, portable radiography may become more widely available, and robotics-supported ultrasound imaging is currently being trialed. Finally, novel blood-based tests evaluating the immune response using “omic-“techniques are underway. This approach, including transcriptomics, metabolomic, proteomics, lipidomics and genomics, is still distant from being translated into POCT formats, but the digital development may rapidly enhance innovation in this field. Despite these significant advances, TB-POCT development and implementation remains challenged by the lack of standard ways to access non-sputum-based samples, the need to differentiate TB infection from disease and to gain acceptance for novel testing strategies specific to the conditions and settings of use.

## 1. Introduction

In recent years, it was estimated that approximately 10 million individuals had tuberculosis (TB) each year, of which almost one-third were not diagnosed or reported. This detection gap has further increased in 2020 as reported cases fell from 7.1 million to 5.8 million, with an estimated 4.1 million cases thought to be unreported [1]. In children, the case detection gap is larger than in other age groups, with up to 69% of missed cases projected to occur in children under 5 years of age [2]. Early diagnosis and easy-to-access diagnostic tools are ways to tackle this considerable detection gap, and these are therefore one of the key components of the END TB Strategy of the World Health Organization (WHO) [3]. Thus, accurate point-of-care tests (POCT) suitable for children represent an unmet but urgent clinical need.

Generally, a POCT aims to facilitate the clinical decision making of the health care provider close to the patient by providing fast and accurate information independent of an access to laboratory facilities. Examples of implemented POCTs for the diagnosis of infectious diseases include rapid diagnostic tests for malaria, human immunodeficiency virus (HIV), Ebola, dengue and most recently severe acute respiratory syndrome coronavirus type 2 (SARS-CoV-2) [4,5]. POCTs may be of particular relevance in resource-limited settings; therefore, the WHO has defined the following ideal criteria to guide POCT development: affordable, sensitive, specific, user-friendly, rapid and robust, equipment free and deliverable to end users summarized in the ASSURED acronym. For children specifically, Keitel el al. proposed a refined list of criteria for an ideal POCT [6] (Table 1). The authors particularly highlight the need to provide clinical benefit and adequate operational technology to the setting in which a POCT is being used [6]. In addition, the need for child-friendly sampling has been advocated for in recent years. One such example is the use of stool samples for the molecular diagnosis of TB, although suboptimal sensitivity remains a challenge.

## 2. Currently Available Point-of-Care Tests (POCT) for Childhood Tuberculosis

### 2.1. Detection of Lipoarabinomannan (LAM) in Urine Using Lateral Flow Assays

Tests based on detection of mycobacterial lipoarabinomannan (LAM) antigen in urine have been in use since 2015. LAM is a constituent glycolipid from *Mycobacterium tuberculosis* (*Mtb*) that is released from metabolically active or degenerating mycobacterial cells into urine. The Alere Determine™ TB LAM Ag (AlereLAM) was the first commercially available assay using lateral flow for LAM detection in TB patients. As the sensitivity of the AlereLAM is limited, the WHO, however, recommends its use mostly in inpatient settings [8]. In addition, it is only recommended as a rule-in test for TB disease in individuals living with HIV infection including adults, adolescents, and children who are seriously ill, defined as having fever above 39 °C, being tachypneic and tachycardic and “unable to walk unaided” [8]. Additionally, the test is recommended for those with advanced HIV disease, defined as CD4 cell count of < 200 cells/mm³. Studies determining the test performance of AlereLAM in children living with HIV have shown widely varying sensitivity ranging from 43–65% and specificity from 57–94% [9].

In recent years, a new LAM detection assay, Fujifilm SILVAMP TB LAM (FujiLAM, Fujifilm, Tokyo, Japan) has been introduced. The improvements include a combination of high-affinity monoclonal antibodies against *M. tuberculosis*-specific LAM epitopes and a silver amplification step to increase the visibility of test and control lines. Studies show better sensitivity in both HIV-infected and uninfected patients than its predecessor [10,11]. The sensitivity of FujiLAM in HIV-uninfected TB outpatients, reported in a multicenter cohort study in Peru and South Africa, was 53%, which was five times higher than that obtained by AlereLAM [11]. In a meta-analysis including HIV-infected adults, the sensitivity in patients with confirmed TB was 71% (95%CI: 59–81%) [12].

A Nigerian study found similar sensitivities of FujiLAM in HIV-infected and -uninfected adults (66 and 70%, respectively), whereas a study in Zambia found a sensitivity of 75% in both HIV-infected and -uninfected groups [13,14].

The biological mechanisms for the possible better performance of LAM assays in HIV-infected adults are not fully understood. It has been speculated that the higher concentration of LAM in the urine of HIV-infected individuals may reflect an increased hematogenous spread of *Mtb* to the kidneys in those patients [15]. In that context, and given the increased likelihood of disseminated TB in children, it was hypothesized that LAM assays may have increased sensitivity in children. However, the sensitivity of FujiLAM in HIV-infected children was approximately 55–60% and therefore lower than in HIV-infected adults [9,16]. Interestingly, the sensitivity in HIV-uninfected children was higher than expected, with 60–65%, and was therefore similar to that reported in HIV-infected adults [16,17]. The reasons behind these observations are speculative, but *Mtb* bacterial load and TB severity may influence the detection of LAM in urine [11,18]. In children specifically, FujiLAM sensitivity increased with more advanced stages of the disease, and the number of positive cases was proportionally higher among underweight and stunted children [17]. This suggests that undernutrition may facilitate bacteremia, increasing the amount of LAM in urine samples and thus the sensitivity of LAM tests [19,20]. These findings show that LAM assays have a potential as adjunctive TB diagnostic tests in children, but further improvements in sensitivity are required to enable their use as rule out tests. Specificity in FujiLAM was higher in children compared to AlereLAM, and both tests show a higher specificity in children older than 2 years, suggesting a robust “rule-in” test for older, hospitalized children living with HIV (8). As HIV-coinfection and malnourishment are risk factors for rapid deterioration, ruling-in tests in these individuals is crucial.

### 2.2. Loop-Mediated Isothermal Amplification (LAMP)

The TB loop-mediated isothermal amplification (LAMP) assay, developed by Eiken Chemical Company Ltd. (Tokyo, Japan), is a molecular diagnostic test that detects *Mtb* complex deoxyribonucleic acid (DNA) in sputum samples. The test was officially endorsed by the WHO in 2016 [21]. The LAMP method obviates the need for thermal cyclers required for traditional polymerase chain reaction assays [22]. Further advantages are that the assay provides a result in about one hour, that it can be read by eye under ultraviolet light, and that it requires only limited training. The main disadvantage is that it lacks the ability to detect drug resistance [23]. Additionally, in contrast to Xpert MTB/RIF, the assay has only been validated for use in sputum samples, so that it cannot be used for gastric aspirate specimen, which are more commonly obtained in younger children. Consequently, since the wide-spread implementation of Xpert MTB/RIF, which detects both the organism and rifampicin-resistance, the assay has only found application in a limited range of settings, mainly in environments that prevent the use of the GeneXpert instrument (e.g., with unreliable electricity, extreme temperatures and humidity or excessive dust). Crucially, the WHO recommendations highlight that implementation of TB-LAMP does not eliminate the need for smear microscopy, as the latter is currently recommended for treatment response monitoring. Similarly, the Xpert MTB/RIF assay is also not considered suitable for monitoring as it does not distinguish between viable and non-viable bacilli.

A systematic review and meta-analysis performed in 2016 found that in adults with culture-confirmed pulmonary TB the performance characteristics of the TB-LAMP assay were very similar to that of the Xpert MTB/RIF assay, with pooled sensitivity and specificity estimates of 78% and 99% versus 81% and 98%, respectively [21]. However, subgroup analyses revealed that the sensitivity of the TB-LAMP assay appears to be substantially lower in HIV-infected TB patients, reducing the assay’s usefulness in regions with high HIV prevalence, as illustrated by an adult study in Uganda that reported an overall sensitivity of only 55% [24]. To date, data on the performance of TB-LAMP in children remain very limited, with fewer than a thousand patients included across all published studies combined [25,26,27]. Nevertheless, the available data indicate that the assay has similar sensitivity and specificity in adult and pediatric setting including young children under the age of 5 years, which qualifies TB-LAMP as a potential tool that overcomes the hampering paucibacillary nature of TB in children.

### 2.3. PCR-Based Point-of-Care Tests 

A large number of studies have evaluated the performance of the “classical” Xpert MTB/RIF assay across various locations and using a broad range of clinical specimens (e.g., respiratory samples, pleural fluid, cerebrospinal fluid, ascitic fluid and lymph node material). A recent, detailed meta-analysis estimated the sensitivity of the MTB/RIF assay performed on sputum in children with pulmonary TB to be 65% with a specificity estimate of 99% [28]. The same meta-analysis, using culture-confirmation as the comparator, estimated the sensitivity of the assay in pulmonary TB with different clinical specimens to be as follows: gastric aspirates 73%, nasopharyngeal aspirates 46%, and stool 62%. In the setting of TB meningitis (analysis of CSF) and lymph node TB (analysis of lymph node material), the sensitivity of the Xpert MTB/RIF assay was estimated to be 54% and 90%, respectively. In 2017, a new generation assay termed Xpert MTB/RIF Ultra, incorporating two different multi-copy amplification targets (IS6110 and IS1081) was introduced. To date, the published data on the performance of Xpert MTB/RIF Ultra in children remain relatively limited [29,30,31]. Compared to microbiological reference standards (that varied between studies), key pediatric studies reported sensitivities ranging from 64% to 75% in sputum samples in children with pulmonary TB; however, the study with the lowest sensitivity estimate used frozen sputum pellets instead of fresh samples, which may have impacted assay performance [29,30,31]. Nevertheless, those results suggest that one quarter of children with pulmonary TB have false-negative results, highlighting the need for further improvements of the sensitivity of this assay to be used as a rule-out test.

Additionally, as children have distinctively lower bacterial load compared to adults, “trace” results for *Mtb* detection are more frequent, which also results in loss of rifampicin-resistance results. Despite its limitations, Xpert testing allows for a more rapid diagnosis awaiting culture confirmation, or it can even substitute culture testing in settings where the latter is not available, which is already a considerable progress. Importantly, Xpert using stool specimen has become of interest, because its collection is easier compared to sputum collection in children. Two systematic reviews and meta-analyses showed a pooled sensitivity of 50% (95% CI 0.44–0.55) and 67% (95% CI, 52–79%) compared with the reference standard of culture and/or respiratory samples using Xpert [32,33]. Despite lower overall sensitivity compared to sputum, its child-friendly collection and preclusion of aerosol-generation is promising, especially in the context of adaptation of sample processing, which can further increase the detection yield [34].

The GeneXpert Omni instrument developed by Cepheid (Sunnyvale, CA, USA) is a portable system designed specifically for POCT use [35]. The Omni runs the same PCR-based Xpert cartridges as the ‘classical’ GeneXpert platforms and was therefore expected to achieve similar accuracy as larger, previous generation instruments. However, the commercialization of GeneXpert Omni has been halted by Cepheid [36]. Another GeneXpert instrument that was designed for the low-resource setting and has subsequently been endorsed by the WHO Global TB Program is the GeneXpert Edge system, which was launched in 2018. It comprises a single module that can be powered by an external battery pack and is operated by a touch screen tablet. In common with the Omni, the Edge instrument can process regular Xpert cartridges, although the need for air-conditioned environmental temperatures limits its use beyond a district laboratory level. Early observations from a study in Brazil, which commenced in 2020 but had to be put on hold because of the COVID-19 pandemic, suggest that the Edge instrument facilitates the implementation of TB testing in remote communities with very limited infrastructure [37]. 

In 2020, a further PCR-based assay was endorsed by the WHO. The Truenat assays, developed by MolBio Diagnostics Pvt. Ltd. (Verna, India), are based on real-time PCR chips that are analyzed on the company’s Truelab instruments. The instruments are compact, portable and can be operated with a battery-pack, making them ideal for use as a POCT. So far, all studies evaluating these assays were performed in India, with some including head-to-head comparisons between Xpert MTB/RIF and Truenat MTB indicating both assays had similar performance characteristics [38]. A large multi-center study led by FIND, which included 1807 adult participants with suspected pulmonary TB from study sites in India, Peru, Ethiopia and Papua New Guinea, published its results recently, reporting overall pooled sensitivity estimates of 73% and 80% for the Truenat MTB and the Truenat MTB Plus assay, respectively [39]. However, the sensitivity of both assays was far lower in smear-negative, culture-positive participants (36% and 47%, respectively), which raises concerns regarding their performance in children, who typically have paucibacillary TB disease. To date, the assays have not been evaluated in a dedicated pediatric study. 

There are several other compact POCT molecular test platforms commercially available that enable the detection of a variety of pathogens, including the Alere q system (Alere Inc., Waltham, MA, USA), ID Now (Abbott Laboratories, Chicago, IL, USA), Revogene (Meridian Bioscience, Cincinnati, OH, USA), Biofire Filmarray (bioMérieux, Marcy-l’étoile, France), Q-POC (QuantuMDx, Newcastle upon Tyne, UK) and Cobas Liat (Roche, Basel, Switzerland), but at present, none of those systems have commercially available assays for the detection of *Mtb*. However, several of those manufacturers are currently working on diagnostic TB assays for their POCT platforms. 

### 2.4. Point of Care Ultrasound (POCUS) Imaging for Extrapulmonary Tuberculosis 

Traditionally, ultrasound imaging has been provided within radiology departments. In contrast, point-of-care ultrasonography (POCUS), is performed by the healthcare provider responsible for frontline clinical decision making. Continuing emergence of POCUS applications availability have been fueled substantially by technological advances. Portable devices, handheld smartphone- or tablet-based, as well as wireless systems, have become available at comparatively low costs. Point of care acquisition and interpretation of ultrasound imaging facilitates accurate and timely diagnosis and improves patient outcome. The advantages of POCUS have also been embraced in pediatric care [40]. Children mostly have favorable scanning conditions and benefit highly from POCUS as a quick and non-invasive imaging modality that is free of ionizing radiation, does not require sedation or anesthesia and reduces the need for referral.

The first TB-focused POCUS application named ‘focused assessment with sonography’ (FASH) for HIV-associated TB aimed at improving diagnosis of HIV-associated extrapulmonary TB in adults in South Africa [41]. The FASH protocol concentrates on detection of extrapulmonary TB including pleuritis, pericarditis, abdominal lymphadenitis, hepatic and splenic micro-abscesses. Following a short training, clinicians applied FASH with high confidence in identifying extrapulmonary TB using ultrasound imaging, and patient management was altered in almost half of the patients undergoing FASH [42]. In adults, FASH has been evaluated in different settings and it has become one of the most frequently applied POCUS modules in clinical care in South Africa [43,44,45,46,47,48].

Experience using POCUS to detect extrapulmonary TB in children is limited. It is expected, however, that the benefit in this patient group is high as the proportion of extrapulmonary TB is relatively high in children [49,50]. A first prospective pediatric evaluation of FASH was conducted in South African children and showed a high and HIV-independent yield of positive findings [51]. Surprisingly one-third of children with suspected pulmonary TB had features of concomitant extrapulmonary TB which included pleural effusion, abdominal lymphadenopathy, or focal splenic lesions. Findings of extrapulmonary TB in POCUS were not associated with HIV infection; however, in children with pulmonary TB, HIV infection was more strongly associated with abdominal lymphadenopathy or splenic lesions. At follow-up, three months after initiation of TB treatment, POCUS findings had resolved in 69% of children. In children with persistent POCUS findings, reasons for inadequate treatment response were identified. On the operational side, good or moderate compliance in 80% of children, acquisition of 94% of intended POCUS views, and a high inter-reader agreement showed that POCUS focused on extrapulmonary TB is a feasible bedside imaging tool in children. One intriguing finding is that POCUS studies show extrapulmonary findings in children that would otherwise have been classified as pulmonary TB only. If extrapulmonary findings are more common than previously recognized, this could be a very powerful tool to allow POCUS findings delineating TB from non-TB pneumonia.

### 2.5. Point of Care Ultrasound (POCUS) Imaging for Pulmonary Tuberculosis 

Chest radiographs play a key role in screening for and diagnosis of childhood TB [52,53,54]. Findings on a chest radiograph may support the clinical suspicion of TB in children by demonstrating hilar or mediastinal lymphadenopathy and parenchymal changes [55]. However, the findings are often non-specific and considerable inter-observer variability demonstrates the difficulties in radiological interpretation, even by trained experts [56]. 

Lung ultrasound imaging is a rapidly emerging imaging modality, and for childhood pneumonia, lung ultrasound imaging shows good diagnostic accuracy and diagnostic performance [57]. Lung ultrasound imaging can visualize correlates of pulmonary TB such as consolidation, cavitation, and miliary TB [58,59,60,61], provided the changes have pleural contact with no intervening aerated lung. Demonstration of mediastinal lymphadenopathy, and to a lesser extent also hilar lymphadenopathy, on ultrasound imaging in children is achieved using suprasternal and parasternal windows [62,63,64,65]. Ultrasound imaging was found to be equivalent to computer tomography and have higher sensitivity than chest radiography in detecting mediastinal lymph nodes [62].

Data on chest-POCUS in children with pulmonary TB are limited to one study from South Africa [66,67]. Clinician-performed bedside scanning followed previously published protocols for anterior, posterior, and lateral chest scanning for childhood pneumonia and suprasternal scanning for mediastinal lymphadenopathy [63,68]. Chest radiography and lung ultrasound performed similarly for detection of large (≥0.5 cm) consolidations; in contrast, small consolidations (<0.5 cm) seen on ultrasound imaging were reported on chest radiography in 28% of cases only. Enlarged lymph nodes were seen in 23% of children by POCUS and in 6% by chest radiography with no difference between children with confirmed, unconfirmed or unlikely pulmonary TB. At follow-up, after three months of treatment, consolidations decreased and resolved less commonly in children with pulmonary TB, compared to children with unlikely TB, and enlarged lymph nodes were no longer seen in the unlikely TB category. The quality of the chest ultrasound images was rated good in 83% of cases; however, mediastinal ultrasound imaging could not be evaluated in 31% of the children because of poor image quality or poor compliance. The inter-reader agreement for sonography was superior to the inter-reader agreement for chest radiography for consolidation, pleural effusion, and mediastinal lymphadenopathy.

In summary, current results of POCUS for pulmonary and extrapulmonary TB suggest the potential for rapid radiological detection, and a beneficial role in monitoring treatment response, although sensitivity and specificity remain to be determined. Currently, experiences with TB-focused POCUS in children appear limited to research cohorts [51,66,67] or individual cases [69]. Feasibility and cost-effectiveness in routine care, in regions with differing TB and HIV prevalence and at various levels of care have not yet been investigated. In low-resource settings, POCUS may contribute to mitigating a lack of specialized healthcare workforce and limited diagnostic infrastructure.

### 2.6. Developments Using Ultrasound Imaging

In a next step, integration of extrapulmonary and pulmonary focused POCUS is important as shown in a multi-center study in sub-Saharan Africa [59]. The improved POCUS protocols also include cut-off sizes for findings to reduce interpretation challenges of minor findings (e.g., minimal free fluid) and include novel signs (i.e., subpleural nodules), recently described as indicative of TB. 

## 3. Pipeline and Developments of Point-of-Care Tests for the Diagnosis of Tuberculosis

### 3.1. Immune Response Biomarkers

The C-reactive protein (CRP), a well-established host immune biomarker of infection, received renewed attention after the WHO recently endorsed CRP as a screening marker in HIV-infected individuals [70]. Advantageous for this approach is that reliable POCT assays detecting CRP are widely available. However, CRP is a sensitive but non-specific marker of infection and inflammation. Consequently, it will not be useful as a stand-alone test for TB evaluation but may be useful for inclusion in a combined biomarker approach. This has been shown in recent promising protein biosignature studies including adults and children [71,72]. 

Among newer host immune biomarkers currently evaluated for TB diagnosis, antibodies, cytokines, ribonucleic acid (RNA) signatures and host cellular markers are the most commonly investigated. A recently published systematic review [73] showed that among 44 biomarkers that met at least one minimal targeted product profile (TPP) defined by the WHO, 37/44 (84%) were host and 7/44 (16%) were pathogen biomarkers. Host biomarkers seem to be sensitive but often have limited specificity. Therefore, they are generally more useful as screening tests rather than confirmatory tests, for which a POCT format is highly desirable. The same review showed that the majority of host biomarkers were multiple marker biosignatures. The outperformance of multiple marker biosignatures over single markers mirrors the complexity of the underlying pathophysiology in TB. Development of multiple marker POCT would thus be a major achievement but poses additional challenges as most POCTs are single marker based. In that context, RNA biomarkers may be particularly attractive, as the analyses of several targets can be readily multiplexed into a single POCT. However, a recent systematic review and meta-analysis showed insufficient accuracy of RNA-based blood signatures for point of care TB confirmation [74]. At this stage, RNA signatures would therefore only reach the TPP targets for a screening test. Another option lies in the measurement of host T cell biomarkers using flow cytometry. Currently, the detection of such markers includes surface molecules such as cluster of differentiation (CD) 27, CD38, or CD153, as well as human leukocyte antigen (HLA)-DR and cell proliferation markers such as Kiel (Ki) 67 [75,76,77,78]. These markers have repeatedly achieved sensitivity and specificity levels compatible with TPP targets for a confirmatory TB test, including in children. Substantial simplification of this approach has been successfully achieved to allow its implementation in a resource-limited setting [79]. Nonetheless, functional T cell assays still require laboratory infrastructure and have a 24 h turn-around time, which remains incompatible with the ambitious POCT TPP target to deliver results on the same day and hampers implementation in remote areas.

Antibodies were the most studied host immune biomarkers. Despite this, older studies evaluating the diagnostic performance of TB-specific antibodies were rather discouraging [80]. However, more recent studies showed promising results for antibody signatures [81,82]. Recently, TB-specific IgG4 has been shown to correlate with disease activity and decrease after treatment [83]. 

The potential advantages of antibody biomarkers include high stability of the marker of interest, exciting robust low-cost assays for other pathogens (e.g., lateral flow assays), and limited operator training. Those features highlight the potential for antibody-based tests to be translated into POCT format.

### 3.2. Biomarker Detection in Urine

Urine is a readily available and non-invasive specimen, and therefore particularly useful for the diagnosis of TB in patients unable to expectorate sputum, such as children, the elderly and adults without a productive cough [13,84]. In addition to the aforementioned AlereLAM and FujiLAM, other rapid urine-based tests for the diagnosis of TB are currently being developed. FLOW-TB is a lateral flow urine LAM assay that includes a concentration step using exclusion-based sample preparation technology to increase the detection of LAM. The assay has shown a sensitivity of 52% in HIV-uninfected patients with a specificity of 87%, which is below the WHO minimum TPP [85]. Along the same lines, a recently published study reported a urine LAM concentration method in combination with a sensitive lateral flow assay to improve LAM detection and showed a sensitivity of 60% and specificity of 80%, which was comparable in HIV-infected and -uninfected patients [86]. In addition, further studies have highlighted the importance of sample pre-treatment to improve LAM assay performance [87].

A recent development is PCR-based detection of TB-specific cell-free DNA fragments excreted in the urine [88]. Using a highly sensitive sequence-specific purification method, the first results of this urine-based assay in a South African cohort showed a sensitivity of 84% and a specificity of 100% [88]. Currently, no data is available from pediatric studies. However, the operational characteristics of this test would need to be simplified to allow its implementation in resource-limited settings.

### 3.3. Artificial Intelligence-Supported Interpretation Radiography

Artificial intelligence (AI)-based interpretation of medical imaging has recently been extensively researched and developed further, and multiple commercial products have become available for clinical use [89,90]. Computer-aided detection (CAD) products indicating the likelihood of TB use AI to analyze radiographs and express abnormality scores. Diagnostic performance of CAD software proved similar to interpretation of digital chest radiography by a human reader, with estimates ranging around 90–92% for sensitivity and 23–79% for specificity [70]. CAD has recently been conditionally recommended by WHO as an alternative to human interpretation of digital chest radiography for screening and triage of TB in individuals ≥ 15 years of age [70]. CAD systems are continuously being improved to further optimize test accuracy, turnaround times (currently < 15 s per chest radiograph) and costs (currently around USD 8 per analysis). Prerequisites for CAD in routine care comprise digital radiography equipment and internet access, as well as funds for the CAD product license fees [70]. Few CAD studies focusing of pneumonia and TB have been conducted in pediatric populations [91,92,93]. Despite the challenges of a higher spectrum of chest shapes and disease presentation in younger children, CAD algorithms can outperform human readers and a first software (CAD4TB v6) has recently been licensed for children ≥ 4 years of age. 

## 4. Potentially Available Point-of-Care Tests in the More Distant Future

### 4.1. Portable Radiography 

Digitalization and development of multi-layer detectors have enabled increased availability of radiography and enhanced the development of portable systems for POCT use. The newest portable systems, called ultra-portable or handheld x-ray systems, come with compacted technology that allows safe TB screening in remote locations or home-based environments [94]. At least three ultra-portable radiography systems are commercially available: MINE 2 (HDT; Gwangju, Korea), Xair (FDR XD2000; Fujifilm Corporation; Tokyo, Japan) and Delft Ultra (Delft Imaging Systems, JD’s-Hertogenbosch, The Netherlands). In addition, two of these include platforms include AI-supported interpretation of images [94]. However, results from a performance and implementation evaluation of such a system, in community-based active TB case finding campaigns, found shortcomings in capture capacity per charge and image quality compared to a stationary radiographic system, but abnormality scores using the AI software did not differ between images by the ultra-portable versus reference system [94]. 

### 4.2. Artificial Intelligence and Robotic Supported Ultrasound Imaging

Research and development into AI for ultrasound imaging is less advanced compared to radiography. Obstacles comprise operator-dependency and lack of large data sets with standardized views. So far, AI endeavors for ultrasound imaging focused on algorithms for respiratory conditions because of the comparably consistent acquisition of images and homogenous sonographic pattern of healthy lungs [95,96,97]. Pilot data on a pediatric lung ultrasound-based AI algorithm, which correctly identified pneumonia infiltrates with 91% sensitivity and 100% specificity [97], indicate a potential future role for AI-based ultrasound image interpretation. Research is further exploring ultrasonography operated by robotic arms which detect target regions automatically and adjust to tissue movements by processing images rapidly whilst maintaining high image quality [98]. An autonomous, hands-free hyper-simplified, pediatric ultrasound device is also being developed; the device comprises a wearable plate with a transducer array and sensor housing in the center and includes machine learning-driven onboard image interpretation for heart or lung abnormalities, which has been selected for inclusion in the WHO’s 2021 Compendium of Innovative Health Technologies for Low Resource Settings [99].

### 4.3. The “Omics” Approach and Digital Development

Advancing TB diagnostics by investigating signatures from “omic” analyses is a promising multi-dimensional and unbiased approach [100]. Transcriptomics, metabolomics, proteomics, lipidomics and genomics therefore provide unique opportunities for high-throughput screening for novel TB diagnostic markers [100]. For example, in 2019, relevant alterations in the expression of several types of small non-coding RNA have been detected in TB infection and disease, suggesting a potential diagnostic role in detecting different TB stages [101]. On the pathogen side many lipids such as trehalose, polyketides, mycolyls are specific for *M. tuberculosis,* and therefore lipidomics may be a further promising way to screen for specific mycobacterial components and diagnose TB [102,103]. Furthermore, proteomics may also be an attractive approach for non-sputum specimens, as shown in a study identifying 26 TB-specific proteins in sweat samples [104]. In addition to highlighting the potential of proteomics, this study also highlights the feasibility of non-invasive sampling. Recently, an association has been identified between the urine metabolomic fingerprints and clinical case definitions used for TB classification in children [105]. Sweat, saliva and exhaled breath all have gathered increasing attention as being accessible non-invasively and providing next-generation biomarkers [106]. 

The “omics” approach and detection of biomarkers is inevitably connected to digital development. The internet of (medical) things—which means medical devices and applications are connected to healthcare systems through online computer networks—provides a continuously growing field in which patients and healthcare workers are connected and machine-to-machine communication allows for the rapid exchange of results [107,108,109,110]. According to the principles for digital development, open-source platforms—being device- and analytic-agnostic—are key to fostering a sustainable digital development [111,112]. By adding smart algorithms and applications to biosensors, patients are directly engaged and empowered to participate in managing their health and disease. These technical improvements further enable accessing of alternative biofluids such as sweat, saliva and breath [108]. Due to their connection to smartphones, no direct contact to healthcare workers is needed, making them ideal POCT. Despite the wearables’ high application potential in pediatrics, most are currently not primarily designed for children and adolescents [113]. 

## 5. Challenges for Tuberculosis Point-of-Care Tests

Despite significant advances in TB point-of-care diagnostics over the past decades (Figure 1), many barriers stand in the way of adequate prospects for children on a community level. Recent POCT developments in TB, including lateral flow assays that detect antigens in urine, imaging and immunodiagnostics have focused on non-sputum-based diagnostic approaches. This is crucial, as sputum expectoration is often not possible in children. Furthermore, the paucibacillary nature of TB in children hampers microbiological confirmation. As children are at higher risk of developing extra-pulmonary TB compared to adults [114], non-sputum-based rapid diagnostics have the potential to detect miliary TB and TB meningitis early, mitigating their devastating consequences [115]. 

None of the POCTs currently in use have focused on TB infection, as the implication of detecting TB infection has until recently been underestimated [116]. Nevertheless, the development of a POCT that has the ability to distinguish between TB disease and TB infection remains a crucial task. Accurate and timely diagnosis of TB infection and subclinical TB is key to limit transmission and to reduce the risk of developing TB disease subsequently, especially in younger children [117]. Identification of and differentiation between TB infection and disease is a prerequisite for accurate management and treatment. Finally, disease control in endemic countries will benefit and rely on from accurate POCTs for TB, which has been seen in the case for HIV and malaria [118,119].

Apart from the challenges due to the complexities of the host and the pathogen, sociopolitical hurdles need to be overcome. POCTs require delivery to remote places with limited infrastructure, and inequities in access to diagnostics also must be addressed. Novel biomarker identification should focus on children, rather than extrapolation from ongoing studies from adults, as the pediatric population presents with distinct clinical picture and has other requirements for adequate testing. 

Whilst much development is still required, the ongoing COVID-19 pandemic shows strikingly that tools to prevent, diagnose and treat a disease can be developed with unprecedented speed, resulting from joint collaborations and prioritization of scientific funding. Notably, in 2020, research and development investment in diagnostics have been nearly ten times higher for COVID-19 compared to TB diagnostics in 2019 [120,121]. Ultimately, only prioritization of childhood TB research and TB science spending on a global scale will lead towards accurate TB POCT diagnostics, which are vital to ending the global TB pandemic. 

Figure 1 Overview of current, future and ‘ideal’ point-of-care tests for the diagnosis of TB in children.

## Figures and Tables

**Figure 1 pathogens-11-00327-f001:**
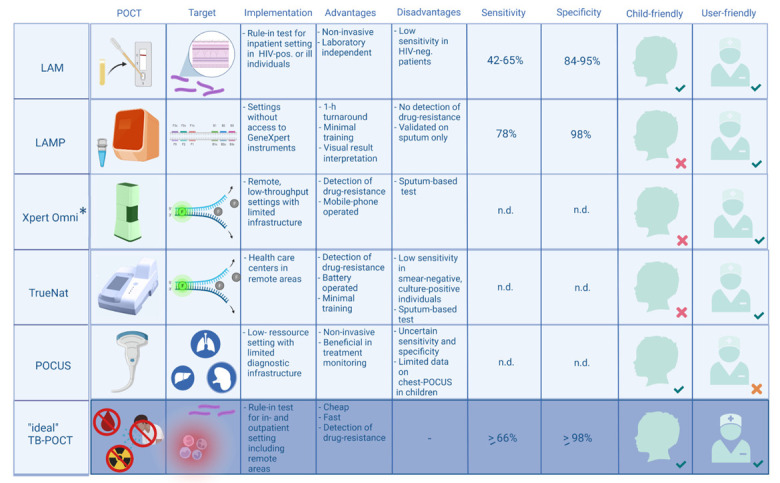
Overview of current, future and ‘ideal’ point-of-care tests for the diagnosis of TB in children. Lipoarabinomannan (LAM) lateral flow assay detecting a component of the cell wall of *M. tuberculosis* in the urine. Sensitivity and specificity refer to PTB in children. TB-loop-mediated isothermal amplification (LAMP), a molecular POCT detecting *M. tuberculosis*. Sensitivity and specificity refer to PTB in adults. Xpert Omni, a PCR-based Xpert cartridge for detecting *M. tuberculosis. * Commercialization of Gene Xpert Omni has been halted by Cepheid in 2021.* TrueNat, based on real-time PCR chips for detecting *M. tuberculosis*. Point-of-care ultrasonography (POCUS) visualizing correlates of parenchymal pulmonary pathology in pulmonary TB as well as abnormalities in other anatomical locations in extra-pulmonary TB. It is only conditionally user-friendly, as it requires minimal skills acquisition. The “ideal” POCT for TB in children is cheap, fast and non-invasive. It should not harm the child via radiation or invasive procedures and be sputum-independent. The target can be based either on the host or on pathogen properties. According to the WHO Target Profile for a rapid biomarker-based non-sputum-based test, it should have at least a sensitivity of >66% and an optimal specificity of more than or equal to 98%. The health care provider should be able to perform the test without extensive training or skills.

**Table 1 pathogens-11-00327-t001:** Currently published and proposed criteria for ‘ideal’ point of care tests in general and for childhood tuberculosis.

Aspects to Be Considered	WHO ASSURED Criteria [7]	For Pediatric Infectious Diseases [6]	Proposed: for Childhood Tuberculosis
**Cost**	Affordable	Cost-effective	Cheap
**Diagnostic performance**	Sensitive	Sufficient diagnostic accuracy and reliability	Sensitive
	Specific	Impact on patient outcome/clinical benefit	Specific
**Usability**	User-friendly	User-friendly	User and child-friendly
**Turnaround time**	Robust and rapid	Rapid	Rapid
**Resources**	Equipment-free	Adequate operational technology geared to environment of application	Applicable in low-resource settings
**Target group**	Deliverable to those who need them		Children in high- and low resource settings
**Specific requirements**			Differentiation between TB infection and disease
**Other**			Information on drug resistance

## Data Availability

Exclude as this statement if the study did not report any data.

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
