# Peer review of "Towards Accurate Point-of-Care Tests for Tuberculosis in Children"

_pathogens, 2022, doi:10.3390/pathogens11030327_

Round 1
Reviewer 1 Report
Overall comments: This is generally a well-written review focusing on the current and future state of point-of-care TB diagnostics in children. There are several sections that would benefit from re-structuring content and some clarifications, as suggested in detail below. An overarching comment is in regards to the manuscripts focus on diagnosis of TB disease versus infection. The review title mentions POCT for both TB infection and disease, yet the bulk of the review is dedicated to diagnosis of TB disease. The review would need to be expanded significantly to capture the nuances of delineating between TB disease and asymptomatic infection; would suggest authors maintain focus on POCT for TB disease in children, and state this focus in the introduction. Title and some wording throughout the review could then be clarified. If the authors would like to include PCOT for TB infection in absence of overt disease, very substantial additions need to be considered. My detailed comments are only in relation to POCT for TB disease.
Abstract:
Overall, the abstract should be more concisely structured. Would consider presenting as: new tools to improve detection of TB in respiratory and gastric aspirate samples (LAMP; Xpert), urine based (LAM), blood based (immune biosignatures; multi-omics), and imaging modalities (US).
Line 31: “Early diagnosis and easy to access diagnostic tools are ways to tackle this considerable detection gap.” I am not sure I understand how early diagnosis can address a problem with diagnosis, seems like very circular logic? I would consider restating this, eg “Easy to access and accurate diagnostic tools are essential to early diagnosis and management of TB disease in children, and will lead to reduced TB-related morbidity and mortality in this vulnerable population.”
Introduction:
Line 65: would suggest change “and laboratory-independent information “ to "fast and accurate information that is not dependent on access to laboratory facilities".
Not sure that Table 1 adds much to what is detailed in proceeding paragraph. All the information in the table seems fairly obvious and already described as text. Authors could consider removing Table 1.
Section 2:
LAM: Can you comment on the specificity of FujiLAM in children (HIV+ and HIV-). If robust, can it be applied/recommended as a "rule in" test for pediatric TB suspects? If the LAM assay performs well in malnourished children that is a significant advancement, given the complex associations between TB and wasting in children. More emphasis on the promising aspects of the assay should be considered (theme throughout review).
LAMP: Has it been validated or tested in gastric aspirates? Would include such info given articles focus on pediatric diagnostics and many centers still rely on GA to obtain samples from younger children
Lines 137-139: “Crucially, the WHO recommendations highlight that implementation of TB-LAMP does not eliminate the need for smear microscopy, as the latter is currently recommended for treatment response monitoring.” Could authors frame this in the context for WHO recommendations for use of AFB smear when GeneXpert is available?
Lines 150-151: If the data suggests similar diagnostic accuracy in peds vs adult TB, isn't that important to emphasize? Every other diagnostic test for TB performs better in adults than kids. So I would frame this as potentially very promising. However, would be important to comment on the age of those children included in current TB LAMP data. Less exciting if only tested in older children who exhibit adult-like disease.
PCR-based point of care assays:
This section would benefit from re-organization, starting with discussion on “classical” GeneXpert, then Ultra, then new GeneXpert systems, then alternative PCR systems. Take the reader from what is more familiar to what is emerging & new.
Lines 153-164: Would be consistent in how refer to 1st Xpert cartridge that was available. Pick one term, "classical" or "initial" but be consistent.
It would be worth adding some additional details on the Ultra platform regarding the "trace" read out, and the difference in how this is interpreted in adults vs children vs PLWH. Should be mentioned that lose RIF testing for samples in the "trace" range (which are many of the peds+).
Indeed correct that Xpert, including Ultra, is still missing a bunch of kids who will go on to have culture-confirmed disease. But, it is also diagnosing a huge # of kids much sooner that waiting for a culture result, and is available in (some) settings where culture does not exist. So given the vast majority of kids are AFB smear negative, Xpert is allowing a much faster diagnosis of TB disease in many children--even if not perfect, this is good progress. Some emphasis on the positive way these tools can be used rather than just naming limitations would be appreciated in the review.
I would incorporate more discussion on the role of stool as a biologic sample for molecular testing in children (particularly using Xpert/Ultra). Very, very limited mention here, yet it could lead to a very promising new approach to Peds TB diagnosis and several good studies published. Limitation of current data (eg. different processing methods of stool), as well as current recommendations, should be included.
Section on US
I would strongly consider moving section 2.5 (US for pulm TB) immediately after line 218, and before the section on extrapulmonary TB. It seems very odd to have the extrapulmonary data presented prior to the pulm data (even if that’s how the studies were done).
Most sections in this review start with a discussion of the adult studies, then into the peds studies. Here it is odd to me that no discussion about POCUS for adult pulm TB is included; this should be considered as an addition.
What is notable about the Lung US studies, I think, is that we are finding many features of extrapulmonary TB in people who would be clinically classified as pulmonary TB. So we are learning that our previous clinical tools to identify extrapulmonary involvement of TB were inadequate. Whether or not this any has impact on treatment decisions and outcomes is to be determined (eg. clinical signs symptoms suggest only pulmonary disease, yet you see abdominal LAD, for example. Does this impact treatment decisions? Outcomes? We don’t know yet). For young children, it will be exciting to seek opportunities to study children with TB pneumonia and those with non-TB pneumonia. One could hypothesize that only children with TB pneumonia will exhibit extrapulmonary findings. If true, this could be a powerful clinical tool to allow US findings support delineation between TB and non-TB PNA in endemic settings. Would be great to see some of these concepts incorporated in this section if authors are in agreement.
Section 3:
I would start this section with the more commonly recognized biomarkers (eg, CRP) and then go into the more novel ones. Better for reader.
Line 311: Functional T cell assays are far beyond the capacity of the vast majority of TB endemic settings. I would say the 24h turn-around time is the least of the limitations as POCT (eg. you still need laboratory infrastructure or the capacity to ship samples very quickly to a reference lab).
Lines 317-319: More detail here is needed, otherwise really confusing given failure of prior AB diagnostics for TB. Functional antibody signatures that examine changes in the biochemical modification of the molecules themselves have led to the breakthroughs in using antibodies to support TB diagnosis. This has only been established in adults, however, and we don't know yet if the same functional ab signatures would be present in pediatric TB (suggest looking at work of Dr. Galit Alter). AB-based diagnostics could also get complicated in infants where maternal IgG still circulating, so this is a specific barrier for their use in younger children (who are also the most at risk).
Section 3.3: I would merge this with the LUS section. It's not a new technique, just a new study protocol. Plus, it nicely brings up the issue of imaging outside the lungs for children with suspected pulmonary TB as a way of increasing diagnostic accuracy.
Paragraph, lines 422-434: I do not understand the central message of this paragraph or relevance to the review. The phrase “internet of things” is particularly problematic.
Section 5:
Lines 437-438: This sentence (although the content is fine) seems very out of place given the rest of the paragraph is positive and discussing the positive aspects of moving away from sputum based diagnostics for children.
Line 446: “Differentiation between active and subclinical forms of TB is another challenge for POCT in TB”. Seems odd to start discussing this important issue here, at the end of the article. This review really only seems to be focusing on diagnosis of TB disease. I would be very clear about that in the introduction, as assays (whether point of care or not) that attempt to discriminate between infection and disease states seems beyond the scope of this review. If authors want to maintain this within the review, please include how they define TB infection versus subclinical TB and if it is important to distinguish.
Additional comments for consideration:
Regarding novel biomarkers for TB disease, the investigations almost universally begin with adult TB patients, find something, and then try to apply to pediatric TB with subsequent loss of diagnostic accuracy. We should consider that biomarkers ID in adult TB patients may be irrelevant for pediatric TB. Perhaps if we want to identify better tools to diagnosis TB in children, we should be doing the discovery work in children too.
Reviewer 2 Report
Dear Authors,
below you can find my comments on your manuscript:
- Introduction, lines 54-62: please mention the reasons for the increased case detection gap in children with TB, in comparison to adults.
- Which of the currently available and pipeline POCTs provides information about drug resistance? Is it only rifampicin-resistance?
- I suggest that you provide a table with all the currently available POCTs, summarizing their characteristics, implementation spectrum, advantages and disadvantages.
- Figure 1. You pointed ultrasound as not user-friendly in figure 1. Is it because training is required to implement this method? Please become more specific in this point.
Round 2
Reviewer 1 Report
The authors have appropriately revised the original manuscript, and this has significantly improved its readability and clarity. There is one section that would benefit from editing, (line 140-144)
"the WHO recommendations highlight that implementation of TB-LAMP does not eliminate the need for smear microscopy, as the latter is currently recommended for treatment response monitoring. Xpert MTB/RIF in contrast is not suitable for monitoring as it does not distinguish between viable and non-viable bacilli.
Here, neither TB-LAMP or Xpert is recommended for monitoring treatment response. To clarify this point, would suggest that 2nd sentence be edited slightly, "Similarly, the Xpert MTB/RIF assay is also not considered suitable for monitoring as it..."
Author Response
We thank the reviewer for the thorough review report in round 1 and have edited the following section as suggested in round 2:
Comment: There is one section that would benefit from editing, (line 140-144)
"the WHO recommendations highlight that implementation of TB-LAMP does not eliminate the need for smear microscopy, as the latter is currently recommended for treatment response monitoring. Xpert MTB/RIF in contrast is not suitable for monitoring as it does not distinguish between viable and non-viable bacilli. Here, neither TB-LAMP or Xpert is recommended for monitoring treatment response. To clarify this point, would suggest that 2nd sentence be edited slightly, "Similarly, the Xpert MTB/RIF assay is also not considered suitable for monitoring as it..."
Answer: as suggested, we edited the sentence as following: "Similarly, the Xpert MTB/RIF assay is also not considered suitable for monitoring as it does not distinguish between viable and non-viable bacilli"
Additionally. we edited the following section in pag. 3 for clarification:
"The sensitivity of FujiLAM in HIV-uninfected TB outpatients reported in a multicenter cohort study in Peru and South Africa was with 53% five times higher than that obtained by AlereLAM [11].In a meta-analysis including HIV-infected adults, the sensitivity in patients with confirmed TB was 71% (95%CI: 59–81%) [12].
A Nigerian study found similar sensitivities of FujiLAM in HIV- infected and -uninfected adults (66 and 70% respectively), whereas a study in Zambia found a sensitivity of 75% in both HIV-infected and -uninfected groups [13,14]."
On behalf of the co-authors,
sincerely,
Nina Vaezipour